# Interplay between the m^6^A Epitranscriptome and Tumor Metabolism: Mechanisms and Therapeutic Implications

**DOI:** 10.3390/biomedicines10102589

**Published:** 2022-10-15

**Authors:** Asa Sergei Fong, Yasi Pan, Jun Yu, Chi Chun Wong

**Affiliations:** Institute of Digestive Disease, Department of Medicine and Therapeutics, State Key Laboratory of Digestive Disease, Li Ka Shing Institute of Health Sciences, CUHK Shenzhen Research Institute, The Chinese University of Hong Kong, Sha Tin, N.T., Hong Kong 999077, China

**Keywords:** N^6^-methyladenosine, cancer, metabolism, glucose, lipid, amino acid

## Abstract

N^6^-methyladenosine (m^6^A) modification of messenger RNA (mRNA) influences the stability and translation of the transcripts into functional proteins. Recent studies reveal the role of m^6^A modifications in regulating the metabolism of basic biomolecules such as glucose, lipids and amino acids. Such mechanisms are not only important for physiological functions of normal cells but also prove to be pivotal for the pathogenesis of cancers by driving dysregulated metabolism. M^6^A writers, readers and erasers function co-operatively to promote aberrant glucose, lipid and amino acid metabolism in cancer cells, which in turn support increased proliferative and metastatic potential. Better understanding of the relationship between m^6^A and metabolism in malignancy may unravel novel therapeutic targets as well as biomarkers in cancer. In this review, we summarize the recent evidence demonstrating the interplay between m^6^A modification and cancer metabolism and their therapeutic implications.

## 1. Introduction

N^6^-methyladenosine (m^6^A) is the most common post-transcriptional modification in messenger RNA (mRNA) of eukaryotic cells, occurring at the N^6^ position of adenosine nucleotides. It is known that an m^6^A-modified transcript can contain multiple modifications at different positions [1]. M^6^A affects mRNA stability and mediates alternate splicing, intron retention and translation. The level of m^6^A modification is tightly regulated by a series of m^6^A modification enzymes which can be classified into “writers”, methyltransferases that install m^6^A modifications, and “erasers”, i.e., demethylases that remove m^6^A from mRNA, as well as “reader” proteins that recognize and bind to m^6^A-modified mRNA to that mediate their ultimate fate, such as promoting its translation to generate proteins or degradation of the transcript (Figure 1). 

M^6^A writers include METTL3, METTL14, Wilms’ tumor 1-associating protein and METTL16. METTL3, METTL14 and WTAP form a complex to catalyze the majority of m^6^A modifications. It was found that WTAP recruits METTL3, while METTL14 mediates binding to mRNA [2]. METTL16 is a recently discovered m^6^A writer of mRNA and U6 small nuclear RNA (snRNA), and it principally regulates S-adenosylmethionine synthetase intron retention, related to methionine metabolism [3]. On the other hand, fat mass and obesity-associated protein (FTO) and ALKBH5 are m^6^A erasers. Both FTO and ALKBH5 catalyze oxidative demethylation of m^6^A in an iron (II)- and α-KG-dependent manner [4,5]. M^6^A readers include the YTH domain proteins [6] as well as insulin-like growth factor 2 binding protein (IGF2BP) family [7]. In the former, YTHDF1 and YTHDF3 induce the translation of m^6^A-modified transcripts [8], while YTHDF2 is conventionally known to bind to m^6^A-modified transcripts and possibly destabilize mRNA, thus causing their premature degradation and downregulation of expression [9,10]. However, studies suggested that YTHDF2 could stabilize MYC and VEGFA transcripts in glioblastoma [10]. The IGF2BP protein family depends on KH domains to bind mRNA, subsequently recruiting RNA stabilizers such as ELAVL1, MATR3 or PABPC1 to promote mRNA stability and translation of m^6^A-modified transcripts [11]. 

It is estimated m^6^A exists in around 20–40% of mRNA transcripts in our body [12]. It can be found in mRNA transcripts of enzymes and proteins involved in metabolism of basic biomolecules, including glucose, lipids and amino acids. This review aims to summarize the involvement of m^6^A modifications in the control of metabolic pathways and their implications in cancer. These basic building blocks of life are not only in constant physiological demand but also contribute significantly to disease pathogenesis, for example cancers, as they are required by actively proliferating tumor cells. Dysregulated m^6^A promotes aberrant metabolism in cancer cells, making them useful markers for therapeutic targeting and prognosis. Thus, understanding the crosstalk between m^6^A and metabolism in promoting cancer have key clinical ramifications.

## 2. m^6^A and Glucose Metabolism in Cancer

### 2.1. m^6^A Writers and Glucose Metabolism

Accumulating evidence suggests that m^6^A writers, in particular METTL3, play a role in promoting glucose turnover in cancer cells. Several studies have documented that level of METTL3 correlates positively with glucose metabolism in cancer patients. Colorectal cancer (CRC) and esophageal carcinoma patients with high METTL3 expression also demonstrated high ^18^F-FDG uptake in tumors indicative of high glucose uptake [13,14]. In accordance with the increased glucose turnover, overexpression of METTL3 is shown to increase the expression of various glycolytic enzymes in tumors including CRC and cervical cancer [13,15,16]. Conversely, knockout of METTL3 cancer cells led to a decline in glucose consumption, along with reduced ATP and lactate synthesis [13,15,16]. These studies shed light on the importance of METTL3 in driving glycolysis in human cancers. 

Several hypotheses have been put forward whereby METTL3 overexpression in cancer promotes glycolysis, including multiple steps involved in glucose uptake and metabolism. GLUT1 is the major glucose transporter upregulated in cancer cells [17]. Chen et al. showed METTL3 increases the translational efficiency of GLUT1 in CRC cells and enhances glucose uptake and lactate production [18], whereas the effect of METTL3 on other glucose transporters such as GLUT2 and GLUT3 is minor. In line with the function of METTL3 as a methyltransferase, METTL3 significantly increases m^6^A modification of GLUT1/SLC2A1 transcript and its half-life, thereby promoting GLUT1 protein expression and glucose uptake. Consequently, GLUT1-mediated lactate could activate mTORC1 cascade and colorectal tumorigenesis. The co-targeting of METTL3 and mTORC1 synergistically suppressed the growth of CRC cells in vitro and in vivo, underscoring the importance of METTL3-driven glucose uptake in CRC [18]. Besides GLUT1, GLUT4 is another glucose transporter reported to be upregulated in presence of METTL3 overexpression [19,20]. 

Glycolysis is a major glucose metabolic pathway upregulated in cancer cells, and METTL3 has been shown to upregulate expression of multiple glycolytic enzymes. Hexokinase, the first enzyme in the glycolytic pathway converting glucose to glucose-6-phosphate, is shown to be upregulated in METTL3 overexpression [13,14]. METTL3-induced m^6^A modification recruits m^6^A readers to stabilize hexokinase 2 (HK2) mRNA at the 3’ untranslated region, thus increasing HK2 expression and promoting the Warburg effect in esophageal carcinoma, CRC and cervical cancer [13,19]. In addition to HK2, METTL3 promotes expression of enolase 2 (ENO2), an enzyme that dehydrates 2-phsophoglycerate to phosphoenolpyruvate, in gastric cancer [21]. Mechanistically, METTL3 mediates m^6^A modification of heparin binding growth factor (HDGF) and stabilizes its mRNA. HDGF in turn functions as a transcriptional regulator that binds to the promoter region of ENO2 and GLUT4 to promote their transcription and expression in gastric cancer cells. 

In addition to increased glycolysis, a hallmark of cancer cells is their preference for lactate production. METTL3 is shown to shunt glycolytic end products to lactic acid fermentation instead of oxidative phosphorylation, hence the Warburg effect. METTL3 promotes m^6^A methylation, mRNA stability and translation of pyruvate dehydrogenase kinase 4 (PDK4) [16]. PDK4 then phosphorylates and suppresses activity of pyruvate dehydrogenase complex, the enzyme responsible for committing pyruvate to the TCA cycle by forming acetyl-CoA. Hence, glycolytic products are denied the entry into TCA cycle, reducing oxidative phosphorylation. Targeted demethylation of PDK4 mRNA by dm^6^A-CRISPR reversed METTL3-induced PDK4 expression and lactate production, implying that METTL3 is a driver of aerobic glycolysis in cancer cells. 

Apart from individual glycolytic enzymes, METTL3 modulates upstream factors that control glycolytic enzyme expression or activity. For instance, METTL3 was found to increase m^6^A of APC mRNA, which promotes its degradation by YTHDF2-mediated mRNA destruction [15]. This relieves the inhibitory effect of APC on β-catenin signaling and its downstream genes CCND1 and c-MYC. The latter upregulates pyruvate kinase M2, catalyzing the final irreversible step of pyruvate formation in glycolysis. In another report, it was shown that METTL3 mediated m^6^A methylation of c-MYC and GLUT1 mRNA during hypoxia stress, a common phenomenon in tumors [20]. Taken together, m^6^A methylation by METTL3 plays an important role in aberrant glucose metabolism in cancer cells.

### 2.2. m^6^A Readers and Glucose Metabolism

Many m^6^A readers are involved in upregulating glycolysis in tumors, often working hand-in-hand with the m^6^A writer METTL3. Nevertheless, the role of readers in glucose metabolism is complex and isoform-specific. M^6^A readers IGF2BP2 and IGF2BP3 raise glucose uptake by inducing glucose transporter expression. IGF2BP2 binds to m^6^A-modified GLUT1 mRNA, and both m^6^A readers stabilize GLUT1/SLC2A1 mRNA in pancreatic cancer and CRC [13]. IGF2BP1 and IGF2BP2 also stabilize m^6^A-modified HK2 transcripts to mediate HK2 protein expression [22]. Similarly, YTHDF1 is recruited to m^6^A-modified HK2 and increases its mRNA stability in cervical cancer, thereby promoting HK2 translation [19]. Emerging evidence suggests that m^6^A readers are mediators of lactate fermentation. In CRC cells, IGF2BP1 binds to 3′UTR of LDHA mRNA, leading to its stabilization and increased lactate dehydrogenase protein levels [22]. Further, IGF2BP3 and YTHDF1 are reported to promote stability of m^6^A-modified PDK4 mRNA, leading to upregulated PDK4 and subsequent downregulated pyruvate dehydrogenase complex [16], thus shuttling pyruvate towards lactate fermentation.

IGF2BP2 is critically involved in signaling cascades that promote glycolysis. IGFBP2 stabilizes c-MYC mRNA to induce c-MYC-mediated transcription of glycolytic genes [23]. It was found that lncRNA LINRIS can directly bind to IGF2BP1 and protect IGF2BP2 from the autophagy-lysosome degradation pathway, thus allowing this reader to promote the expression of glucose transporters and glycolytic enzymes in tumor cells [23]. Conversely, other lncRNAs have been shown to suppress IGF2BP2. FGF13-AS1 binds IGFBP2 and disrupts interaction between IGF2BP2 and c-MYC mRNA, thereby reducing the half-life of c-MYC mRNA and downstream glycolysis activation [24]. On the other hand, LINC00261 was found to reduce c-MYC expression by sequestration of IGF2BP2 [25]. Deregulation of IGF2PB2 thus drives glucose metabolism in cancer. 

On the contrary, the m^6^A reader YTHDC1 functions largely to suppress glucose metabolism [26]. In pancreatic cancer cells, YTHDC1 was shown to downregulate pri-miR-30d and upregulate mature miR-30d in m^6^A-dependent manner by prompting the degradation and maturation of this miRNA. MiR-30d, a glycolysis-associated miRNA, subsequently inhibits the expression of RUNX1, a transcription factor that binds to and promotes expression of HK1 and GLUT1. Correspondingly, reduced HK1 and GLUT1 result in the suppression of aerobic glycolysis and pancreatic tumorigenesis. In another study, YTHDF2 functions as a repressor of glycolysis under hypoxia conditions by promoting the degradation of m^6^A-modified aldolase A (ALDOA) mRNA, a key enzyme in glycolysis [27]. Depletion of ALDOA leads to depletion of lactate and attenuation of tumor growth, whilst knockdown of YTHDF2 rescues ALDOA expression in FTO-silenced HCC cells, implying its role in posttranscriptional regulation of ALDOA under hypoxia conditions. Hence, m^6^A readers could promote or suppress glucose metabolism in cancers in an isoform- and context-dependent manner.

### 2.3. m^6^A Erasers and Glucose Metabolism

FTO, a major m^6^A eraser, is highly engaged in glucose metabolism. In a subset of acute myeloid leukemia, FTO-mediated m^6^A demethylation of phosphofructokinase (PFK) and lactate dehydrogenase (LDH) B mRNAs prevents their decay mediated by m^6^A reader YTHDF2. These leukemic cells are sensitive to the antitumor metabolite, R-2-hydroxyglutarate (R2-HG), which directly suppresses the catalytic activity of FTO, therefore abrogating the FTO/YTHDF1/PFK/LDHB axis and impairing glycolysis [28]. In contrast, in lung carcinoma patients, downregulation of FTO induced by β-catenin contributes to glycolysis [29]. The binding of EZH2 to β-catenin and localization of this protein complex to FTO promoters mediates H3K27me3 and transcription repression. As a consequence, downregulated FTO leads to elevated m^6^A modification of multiple metabolic genes, such as MYC mRNA, and facilitates their recognition and translation by m^6^A reader YTHDF1. This culminates in c-MYC activation, increased glucose turnover and lactate synthesis, thus promoting tumor cell proliferation and metastasis. Most importantly, FTO-dependent m^6^A demethylation could regulate glycolytic enzymes in opposite directions in a cancer-type-specific manner. 

Another m^6^A eraser, ALKBH5, has been shown to demethylate and inhibit the expression of PDK4. As PDK4 mRNA is also a target of m^6^A writer METTL3, ALKBH5 could antagonize lactic acid synthesis in tumor cells elicited by METTL3 [16]. Yu et al. [30] also reported that ALKBH5 is negatively correlated with the expression of glycolytic enzymes such as GLUTs, LDHA and LDHB in bladder cancer cells. Mechanistically, ALKBH5 suppresses stability of casein kinase 2α (CK2) mRNA by binding to its 3′UTR, thus suppressing expression of downstream glycolytic enzymes. Interestingly, a recent study revealed that ALKBH5 targets the lactate transporter (MCT4/SLC16A3) mRNA for demethylation to promote its mRNA stability and expression [31]. Increased MCT4 drives the export of lactate into the tumor microenvironment, leading to inactivation of cytotoxic T-cells concomitant with induction of T_reg_. Co-targeting of ALKBH5 with anti-PD1 synergistically impaired tumorigenesis, highlighting the potential role of ALKBH5 as a therapeutic targeting in immune checkpoint therapy. The overall interaction of glucose metabolism and m^6^A regulators is summarized in Figure 2. 

## 3. m^6^A and Lipid Metabolism in Cancer

### 3.1. m^6^A Writers and Lipid Metabolism

M^6^A writers METTL3 and METTL14 have been shown to promote physiological adipogenesis by increasing CCNA2 m^6^A and expression in adipocytes [32]. Aberrant expression of m6A writers in cancer cells is frequently associated upregulation of lipid metabolism. Zuo et al. discovered that METTL3 stabilizes long non-coding RNA (lncRNA) LINC00958 in hepatocellular carcinoma (HCC) cells via m^6^A modification [33]. Induced LINC00958 acts via the miR-3619-5p/HGDF pathway to increase expression of numerous lipogenic enzymes and proteins, including sterol regulatory element-binding protein 1 (SREBP1), the master transcription factor for fatty acid synthesis, and its downstream targets fatty acid synthase (FASN), stearoyl-CoA desaturase-1 (SCAD1) and acetyl-CoA carboxylase (ACC), leading to exacerbated formation of lipid droplets. A similar phenomenon is observed in the livers of type 2 diabetes mellitus patients [34], where METTL3 drives upregulation of FASN, a rate limiting enzyme in fatty acid biosynthesis. Knockdown of METTL3 thus suppressed fatty acid synthesis. METTL3 has also been reported to promote m6A modification of peroxisome-proliferator-activated receptors (PPARαs) in liver cancer cells, leading to subsequent mRNA degradation in a YTHDF2-dependent manner [35]. PPARαs play a critical role in sensing of fatty acids, concomitantly promoting fatty acid catabolism and suppressing biosynthesis in response to excess fatty acid accumulation. METTL3-mediated PPARα depletion thus causes excess lipid accumulation in HepG2 cells. Aside from promoting fatty acid biosynthesis, METTL3-mediated m^6^A inhibits fatty acid catabolism. Carboxylesterase 2, an enzyme that catalyzes triglyceride hydrolysis, is negatively regulated by m^6^A [36]. Together, m^6^A writers positively regulate lipogenesis but impair lipolysis, particularly in liver cancer. 

### 3.2. m^6^A Readers and Lipid Metabolism

YTHDF2 is the major mediator of lipid metabolism by recognizing m^6^A-modified mRNA transcripts and targeting them for degradation. Lipogenic mRNA such as FASN is reported to be a target for YTHDF2 [37]. Another m^6^A reader, YTHDC2, degrades other lipogenic genes including SCAD1, SREBP1-c and ACC1 [38]. Concordant with the inhibitory role of YTHDF2 in lipogenesis, FTO, a lipogenic eraser, has been shown to downregulate YTHDF2 [37,39]. In contrast, the degradation of PPAR mRNA by YTHDF2 plays a part in reactive-oxidative-species-induced lipogenesis in the liver [35]. Therefore, the role of m^6^A readers in lipid metabolism may vary in a cell- and context-dependent manner.

### 3.3. m^6^A Erasers and Lipid Metabolism

FTO is a major m^6^A eraser involved in the regulation of lipid metabolism. In various settings, FTO is a prerequisite factor for adipogenesis [40,41,42]. In cancer cells, FTO is also important for the expression of lipogenic enzymes. Sun et al. reported increased FASN expression by FTO in HepG2 cells. On the other hand, FASN, ACC and ACLY expression decreased in FTO-knockout cells, leading to impaired lipogenesis and the induction of apoptosis in HepG2 cells [37]. Mechanistically, FTO-m^6^A demethylation of these transcripts prevents their recognition and degradation by YTHDF2. FTO may also increase the expression of lipogenic enzymes by upregulating their transcription factor, SREBP1c [43]. FTO promoted SREBP1c nuclear translocation and maturation, thus improving its transcriptional activity. Mutant FTO deficient in demethylase activity has reduced capacity to promote lipogenic gene expression, underscoring the role of FTO-mediated m^6^A demethylation in promoting lipid metabolism. The overall interaction of lipid metabolism and m^6^A regulators is summarized in Figure 3.

## 4. m^6^A and Amino Acid Metabolism in Cancer

Recent evidence provides insights into the role of m^6^A modifications involved in the metabolism of amino acids, in particular methionine and glutamine. These amino acids are important building blocks for cell growth, development and proliferation, and cancer cells frequently utilize these two amino acids. In this section, we highlight the interplay of m^6^A regulators and the metabolism of these two amino acids (Figure 4).

### 4.1. m^6^A Writers and Methionine Metabolism

Methionine is an essential amino acid involved in multiple biological functions. Intriguingly, a two-way interplay is involved between methionine and m^6^A regulators. As a methyl donor, methionine is critically involved in S-adenosylmethionine synthesis (SAM), a major methylation donor required for protein, lipid, DNA and RNA methylation [44]. Thus, the supply of methionine directly impacts methylation reactions, including that of m^6^A. Villa et al. [45,46] demonstrated the mechanistic target of rapamycin complex 1 (mTORC1) to be involved in SAM and m^6^A-dependent protein biosynthesis to promote cell growth. mTORC1 positively regulates methionine adenosyl transferase 2α (MAT2A) via transcription factor c-Myc to upregulate SAM synthesis, which donates a methyl group to m^6^A writers METTL13 and METTL14. Moreover, MAT2A is essential for the function of m^6^A writers, as the increased translation due to METTL3 and METTL14 overexpression was abrogated by MAT2A knockout and subsequent depletion of SAM. In addition, mTORC1 elevates WTAP expression to promote the formation of METTL3-METTL14-WTAP complex. Interplay between methionine, SAM and METTL3 in inducing cell proliferation has also been documented in other diseases [47], suggesting that methionine-mediated SAM biosynthesis is critical for supporting methylation activities of m^6^A writers. Dietary methionine restriction, along with targeted therapy towards the methionine cycle, might abolish the oncogenic potential of m^6^A writers. 

MAT2A expression is reciprocally controlled by m^6^A modification. MAT2A m^6^A is uniquely catalyzed by METTL16, which methylates in a subset of mRNAs involved in SAM homeostasis. Moreover, METTL16 directly binds to MAT2A mRNA to modulate its processing. Pendleton et al. [3] observed that methionine deficiency promotes the occupancy of METTL16 on hairpin 1 (hp1) of MAT2A 3′UTR, followed by the efficient splicing of retained intron 1 and formation of mature MAT2A mRNA, hence increasing MAT2A protein abundance. On the other hand, in the presence of abundant SAM and methionine, METTL16 catalyzes MAT2A mRNA m^6^A methylation, and it dissociates to favor intron retention. Shima et al. [48] also reported stabilization of MAT2A mRNA when SAM synthesis was inhibited. In support of Pendleton et al., Shima discovered METTL16, rather than METTL3/METTL14, to be the m^6^A writer of the MAT2A 3′UTR region. They further identified YTHDC1 as the reader of m^6^A-modified MAT2A mRNA that accelerates the degradation of MAT2A mRNA. Correspondingly, co-knockdown of METTL16 and YTHDC1 abolished the regulation of MAT2A by varying SAM levels. Another study in C. elegans also identified METTL10, a METTL16 orthologue, as an m^6^A writer for SAM synthetase that is responsive to endogenous SAM levels, implying that m^6^A-mediated regulation of SAM is highly conserved [49]. 

### 4.2. m^6^A Readers and Glutamine Metabolism

Glutamine is an important amino acid for cancer cells, the metabolism of which is found to be related to m^6^A modifications. Cancer cells are addicted to glutamine, which drives the TCA cycle, nucleotide biosynthesis and antioxidative defense [50]. Targeting of glutaminolysis might thus have therapeutic value in cancer [51,52]. Current studies point to the potential involvement of m^6^A readers in glutamine metabolism. YTHDF1 is highly upregulated in colorectal cancer (CRC) via gene amplification [53]. YTHDF1 was reported to bind to the 3′UTR of glutaminase 1 (GLS1) mRNA and promotes the translation of GLS1 protein [54]. The depletion of YTHDF1 subsequently suppresses glutamine metabolism and renders CRC cells vulnerable to cisplatin treatment. This suggests that inhibition of glutamine metabolism via YTHDF1 has a synergistic effect with cisplatin, allowing lower doses of cisplatin and reduction in toxic side effects. Cancer cells respond to glutaminolysis blockade via upregulation of activating transcription factor 4 (ATF4), which increases protein catabolism and re-cycling of amino acids. Han et al. reported that ATF4 provoked pro-survival autophagy in CRC cells upon glutaminolysis inhibition [55]. Importantly, it was shown that the blockade of glutamine metabolism abrogated ATF4 m^6^A modification, thereby preventing YTHDF2-medaited mRNA decay. As a consequence, upregulation of ATF4 induced transcription DDIT4 to block mTOR, eliciting protective autophagy and cell survival. The co-targeting of glutaminolysis and ATF4 is synergistic in suppressing CRC growth, inferring a strategy for cancer treatment.

### 4.3. m^6^A Erasers and Glutamine Metabolism

Among m^6^A erasers, FTO has been associated with glutamine metabolism in cancer. Hypoxia rewires glutamine metabolism towards reductive metabolism, which is essential for biosynthesis of citrate required for lipid metabolism [56]. In renal clear cell carcinoma (ccRCC), loss of the von Hippel–Lindau (VHL) tumor suppressor causes upregulation of hypoxia signaling via HIF1 and HIF2. Xiao et al. [57] identified a novel synthetic lethal interaction between VHL loss and FTO in ccRCC, implying that FTO is essential for cells with activated hypoxia signaling. Integrative m6A-seq and RNA-seq revealed SLC1A5, a glutamine transporter, as a molecular target of FTO in ccRCC cells. Mechanistically, FTO represses m6A modification of SLC1A5 mRNA and promotes its mRNA expression and translation, leading to increased glutamine uptake in RCC and enhanced tumor survival and progression, especially VHL-deficient cells. The loss of FTO thus compromised the capacity of ccRCC cells to uptake and metabolize glutamine, leading to cell death. In addition, FTO is critical to the adaptive response of cancer cells to glutamine deprivation [55]. FTO demethylates ATF4 mRNA so that the latter is less susceptible to YTHDF2-mediated degradation. Upregulated ATF4 in turn promotes survival through the induction of autophagy. In short, FTO is a major player in m^6^A modification of glutamine-related enzymes and proteins in conjunction with m^6^A readers.

## 5. Therapeutic Implications and Future Perspectives

The m^6^A writer METTL3 is an emerging therapeutic target for cancer treatment [18,58]. As summarized above, METTL3 plays an important role in metabolic rewiring in tumors. METTL3 potentiates multiple steps of tumor glucose metabolism, including glucose uptake, glycolysis and lactate production, thereby contributing to the Warburg phenomenon. As a consequence, genetic and pharmacological blockade of METTL3 has shown great promise in the suppression of tumorigenesis. On the other hand, the targeting of m6A readers is more complicated, as different isoforms have contrasting impact on tumor metabolism in a context- and cell-type-dependent fashion. Among m^6^A readers, IGFBP2 is a promising metabolic target, as it has been shown to promote the increased stability of glycolytic enzymes. YTHDF1 might be a useful target to suppress aberrant glutamine metabolism, given its role in promoting the translation of GLS1. In contrast, other m^6^A readers including YTHDC1 and YTHDF2 primarily function as repressors of tumor metabolism. As for m^6^A erasers, FTO is an attractive therapeutic target, as it has been reported to promote glucose, lipid and glutamine metabolism in cancer cells. Several FTO antagonists have been reported in the literature, including rhelic acid [59], meclofenamate sodium [60] and FB23 [61], and it would be of great interest to test whether these FTO blockers could reverse metabolic rewiring in cancer cells (Figure 5). 

In recent years, there has been significant interest in the m^6^A epitranscriptome. Nevertheless, major gaps remain in our understanding of its exact role in regulating metabolism in cancer. Tumor metabolism in conventional cell cultures is often poorly representative of the three-dimensional tumor microenvironment with heterogeneity with respect to nutrient access, oxygen tension and interaction with non-cancerous cells (e.g., fibroblasts, immune cells). Thus far, very few attempts have been made to profile intratumoral heterogeneity of m^6^A. Future studies with the parallel characterization of the epitranscriptome and metabolomic features from multiple sites in a single tumor will offer additional insights into the dynamic regulation of tumor metabolism by m^6^A. Reciprocal regulation of m^6^A modification enzymes by tumor metabolites is also largely unclear. A better understanding of crosstalk between m^6^A and metabolism in cancer will ultimately contribute to the development of novel therapeutic targets and molecular biomarkers for cancer diagnostics/prognostics and treatment.

## Figures and Tables

**Figure 1 biomedicines-10-02589-f001:**
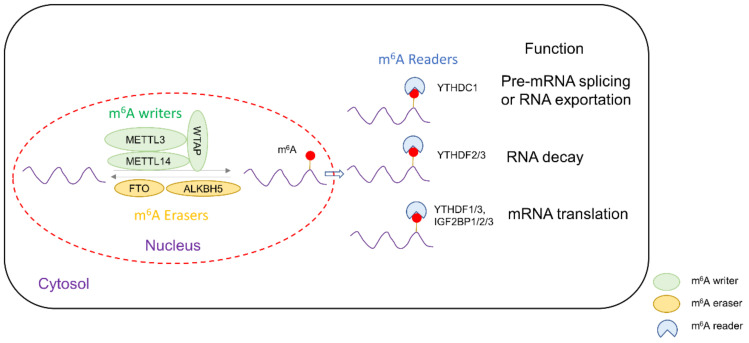
Schematic diagram showing the roles of m^6^A writer, easers and readers in the post-transcriptional regulation of mRNA.

**Figure 2 biomedicines-10-02589-f002:**
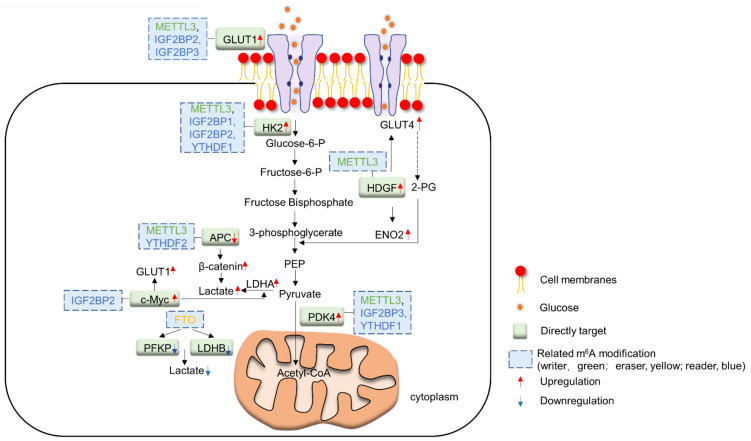
Schematic diagram summarizing the interplay between m^6^A regulators and glucose metabolism.

**Figure 3 biomedicines-10-02589-f003:**
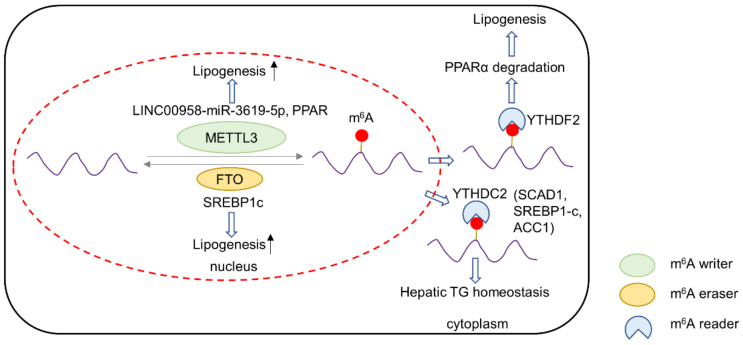
Schematic diagram summarizing the interplay between m^6^A regulators and lipid metabolism.

**Figure 4 biomedicines-10-02589-f004:**
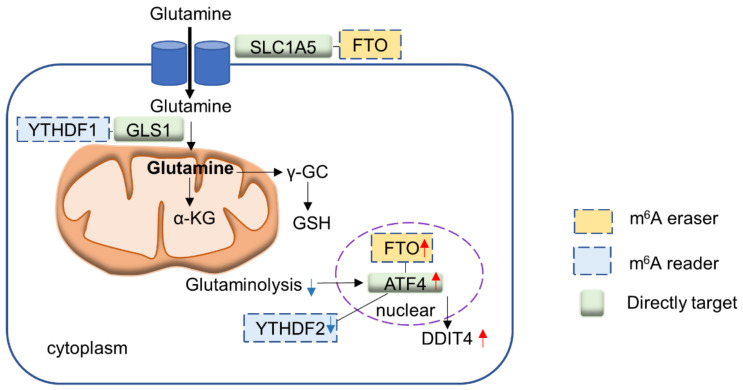
Schematic diagram summarizing the interplay between m^6^A regulators and amino acid metabolism.

**Figure 5 biomedicines-10-02589-f005:**
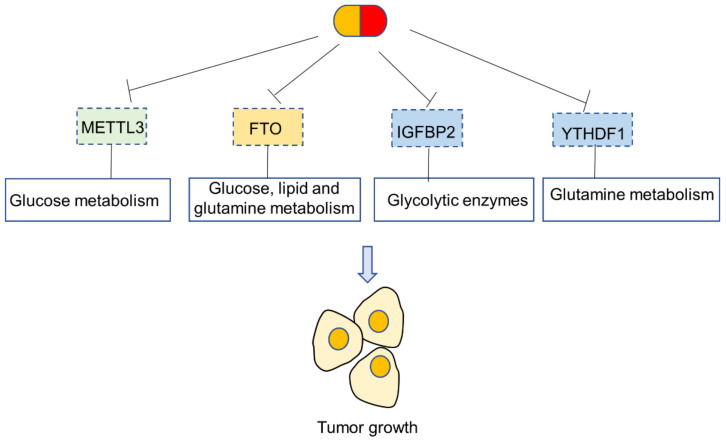
Therapeutic opportunities for targeting m^6^A regulators to reverse metabolic rewiring in cancer.

## Data Availability

Not applicable.

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
