# Peer review of "Interplay between the m6A Epitranscriptome and Tumor Metabolism: Mechanisms and Therapeutic Implications"

_biomedicines, 2022, doi:10.3390/biomedicines10102589_

Round 1

Reviewer 1 Report

Well written review manuscript for the m6A related to glucose/lipid/amino acid metabolism in tumor. Some suggestions:

1. A schematic figure to show the N6-methyladenosine modification to the mRNA is highly recommended for readers curious about details of the m6A.

2. Line 44, instead of leaving readers confused, why not add something like “possibly destabilize the mRNAs expressing the suppressors for those mRNA or with other unknown mechanisms”?

3. For all figures, recommend to make the figure more comprehensive by marking the proteins with colors or tags to indicate their roles to m6A.

4. Figure 2, why double strand?

5. Would be great if there’s a figure for the therapeutic implications

Author Response

We thank the review for the positive comments. Please find our response as follows:

1. A schematic figure to show the N6-methyladenosine modification to the mRNA is highly recommended for readers curious about details of the m6A.

Response: We have added Figure 1 outlining m6A modification in mRNA.

2. Line 44, instead of leaving readers confused, why not add something like “possibly destabilize the mRNAs expressing the suppressors for those mRNA or with other unknown mechanisms”?

Response: We have revised as follows: while YTHDF2 is conventionally known to binds to m6A-modified transcripts and possibly destabilize mRNA, thus causing their premature degradation and down-regulation of expression.

3. For all figures, recommend to make the figure more comprehensive by marking the proteins with colors or tags to indicate their roles to m6A.

Response: We have revised as suggested.

4. Figure 2, why double strand?

Response: We have revised to single strand as suggested.

5. Would be great if there’s a figure for the therapeutic implications.

Response: We have added figure 5 for therapeutic implications as suggested.

Reviewer 2 Report

This review article is covering some aspects of the relationship between m6A epitranscriptome and tumor metabolism in malignancy and their therapeutic implications.

The specific aims of this article are exclusively directed on N6-methyladenosine post-transcriptional modification in messenger RNA of eukaryotic cells targeting adenosine nucleotides.

This will constitute the important goals and novelty of this paper. The article is concluded with a collection of 61 mostly recent references. Additionally, all 3 figures are very informative and with concise important data comparison summarizing the interplay between mregulators and glucose metabolism, lipid metabolism and aminoacid metabolism. 

            The following suggested changes and recommendations should be introduced before the publication of the manuscript.

1.     Page 1. Abstract.  Line 17. Replace “druggable” with “therapeutic” 

2.     Page 4. Line 170. Replace “In other words” with “Most importantly” 

3.     Page 5. Line 209. Replace “Apart” with “Aside” 

4.     Page 7. Line 242. Replace “addicted“ with “ utilizing” 

5.     Page 7. Line 291. Replace “in” with  “of” 

6.     Page 8. Line 336. Replace “explosion in the“ with “enormous“

The manuscript is of good quality and importance and is sequentially written and edited in order to meet the standard for the articles published in Biomedicines. Thus, I certainly recommend it for publication after the correction of these suggested minor changes and recommendations. 

Author Response

We thank the reviewer for the positive comments. Our response is as follows:

1. Page 1. Abstract. Line 17. Replace “druggable” with “therapeutic”

Response: Revised as suggested.

2. Page 4. Line 170. Replace “In other words” with “Most importantly”

Response: Revised as suggested.

3. Page 5. Line 209. Replace “Apart” with “Aside”

Response: Revised as suggested.

4. Page 7. Line 242. Replace “addicted“ with “ utilizing”

Response: Revised as suggested.

5. Page 7. Line 291. Replace “in” with “of”

Response: Revised as suggested.

6. Page 8. Line 336. Replace “explosion in the“ with “enormous“

Response: Revised as suggested.

Round 2

Reviewer 1 Report

Thanks for the update! Now more friendly to readers.